# Discovery of Biofilm-Inhibiting Compounds to Enhance Antibiotic Effectiveness Against *M. abscessus* Infections

**DOI:** 10.3390/ph18020225

**Published:** 2025-02-07

**Authors:** Elizaveta Dzalamidze, Mylene Gorzynski, Rebecca Vande Voorde, Dylan Nelson, Lia Danelishvili

**Affiliations:** 1Department of Biomedical Sciences, Carlson College of Veterinary Medicine, Oregon State University, Corvallis, OR 97331, USA; 2Department of Pharmaceutical Sciences, College of Pharmacy, Oregon State University, Corvallis, OR 97331, USA; 3Department of Microbiology, College of Science, Oregon State University, Corvallis, OR 97331, USA

**Keywords:** *Mycobacterium abscessus*, nontuberculous mycobacteria, NTM, high-throughput screening, small molecules, biofilm, intracellular killing, macrophage

## Abstract

**Background/Objectives**: *Mycobacterium abscessus* (MAB) is a highly resilient pathogen that causes difficult-to-treat pulmonary infections, particularly in individuals with cystic fibrosis (CF) and other underlying conditions. Its ability to form robust biofilms within the CF lung environment is a major factor contributing to its resistance to antibiotics and evasion of the host immune response, making conventional treatments largely ineffective. These biofilms, encased in an extracellular matrix, enhance drug tolerance and facilitate metabolic adaptations in hypoxic conditions, driving the bacteria into a persistent, non-replicative state that further exacerbates antimicrobial resistance. Treatment options remain limited, with multidrug regimens showing low success rates, highlighting the urgent need for more effective therapeutic strategies. **Methods**: In this study, we employed artificial sputum media to simulate the CF lung environment and conducted high-throughput screening of 24,000 compounds from diverse chemical libraries to identify inhibitors of MAB biofilm formation, using the Crystal Violet (CV) assay. **Results**: The screen established 17 hits with ≥30% biofilm inhibitory activity in mycobacteria. Six of these compounds inhibited MAB biofilm formation by over 60%, disrupted established biofilms by ≥40%, and significantly impaired bacterial viability within the biofilms, as confirmed by reduced CFU counts. In conformational assays, select compounds showed potent inhibitory activity in biofilms formed by clinical isolates of both MAB and *Mycobacterium avium* subsp. *hominissuis* (MAH). Key compounds, including ethacridine, phenothiazine, and fluorene derivatives, demonstrated potent activity against pre- and post-biofilm conditions, enhanced antibiotic efficacy, and reduced intracellular bacterial loads in macrophages. **Conclusions**: This study results underscore the potential of these compounds to target biofilm-associated resistance mechanisms, making them valuable candidates for use as adjuncts to existing therapies. These findings also emphasize the need for further investigations, including the initiation of a medicinal chemistry campaign to leverage structure–activity relationship studies and optimize the biological activity of these underexplored class of compounds against nontuberculous mycobacterial (NTM) strains.

## 1. Introduction

*Mycobacterium abscessus* belongs to the group of clinically important nontuberculous mycobacteria that is associated with infections in both immunocompromised and healthy individuals [1,2]. MAB is a rapidly growing environmental organism freely found in soil, dust, and moving and still bodies of water [3] and causes severe lung infections in CF patients or individuals with other underlying pulmonary diseases such as chronic obstructive pulmonary disease (COPD) or bronchiectasis [4]. While MAB is a commonly isolated organism from respiratory specimens, it can also lead to a wide range of infections in the skin, post-surgery or post-trauma soft tissues, the central nervous system, and bacteremia [5,6,7]. The incidence of MAB lung infections has been steadily increasing over the past 25 years [1,8,9,10], often leading to chronic and, in many cases, fatal disease due to the natural resistance of this pathogen to most antibiotics [1].

The ability of MAB to form biofilms in the bronchial airways of patients from which invasive variants emerge is an important aspect of MAB pathogenesis and contributes to the clinical manifestations of infection [9]. MAB forms significantly more and robust extracellular biofilm matrices in synthetic cystic fibrosis sputum medium (SCFM), suggesting a favorable host environmental condition for bacteria to establish biofilms [11] while minimizing the efficacy of antibiotics [12,13,14]. The nature and structure of biofilms, including nutrient and oxygen availability to bacteria, contributes to intrinsic and acquired resistance in mycobacteria. For example, in nutrient-depleted areas in biofilms, bacteria enter a stationary phase that make pathogens insensible to antibiotics [12]. Also, the heterogeneous metabolic states of bacteria within biofilm matrices stimulate variation in the antibiotic susceptibility of facultative anaerobes such as mycobacteria and can lead to the failure of antibiotic treatment [15,16,17]. Mycobacterial biofilms are composed of bacterial cells, exopolysaccharides, glycopeptidolipids, eDNA, proteins, carbohydrates, free mycolic acids, and other surrounding materials which can vary depending on location [18]. Biofilms protect bacteria against the host immune system by impairing the activation of phagocytes and the complement system as well as promoting premature TNF-α-dependent apoptosis in phagocytes [19], while stimulating better invasion through airway epithelial cells [20]. Mature biofilms can prevent phagocytosis [19], but microaggregates with pre-biofilm phenotypes have been demonstrated to increase the bacterial uptake process [21]. Furthermore, biofilms create a physical barrier, increasing resistance against conventional antibiotics by up to 1000 folds [22]. The disruption of biofilm eDNA by treating mycobacterial matrices with DNase I has been reported to reduce established biofilms *in vitro*, while also diminishing mycobacterial tolerance to antimicrobials and improving treatment outcomes with various antibiotics [23].

Furthermore, intercellular signaling through the quorum-sensing phenomenon has a critical role in biofilm matrices’ structural integrity and regulates the bacterial division of labor, metabolic shift, and cooperative activities between cells [24,25]. This mutual interaction adds another level of complexity to biofilms and, in a mammalian host, benefits to pathogen colonization, increased virulence, and adaptations with varied host environments. The infections that emerge from biofilms are commonly resistant to the highest deliverable levels of antibiotics and cannot be treated with standard antibacterial strategies [15,16,26], often resulting in tissue damage and chronic suffering in patients.

Current strategies to combat bacterial biofilms are very limited and antibiotic treatment remains the only effective measure to control biofilm infections in NTM patients. Although many antimicrobial molecules and substances have been discovered and shown to effectively inhibit biofilms and eliminate viable bacteria *in vitro* [27,28], eradicating biofilm infections in vivo remains challenging due to the higher doses required, which often lead to increased toxicity and side effects. In recent years, new concepts have emerged to control bacterial biofilms by disturbing bacterial quorum-sensing factors [29], targeting c-di-GMP nucleotide signaling which is implicated in the pathogenicity-like quorum-sensing process, and inhibiting bacterial amyloids, helping pathogens in their adhesion to surfaces [30,31]. These studies highlight the importance of targeting relevant bacterial factors that are expressed in biofilm conditions and enable pathogens to become phenotypically resistant to antimicrobial treatment [15,16,32,33].

While numerous high-throughput compound screening assays have been conducted, leading to the identification of various classes of antibacterial compounds through the use of mycobacteria grown *in vitro* in diverse culture media and conditions [34,35], this approach often overlooks compounds that might target bacterial factors exclusively expressed during the physiological stages of biofilm formation or biofilm maintenance. Targeting these factors may have a significant impact on mycobacterial biofilms, which are difficult to eliminate due to their enhanced resistance to antibiotics and inherent persistence to host immune responses. Consequently, environments that more closely mimic *in vivo* conditions, such as synthetic cystic fibrosis sputum media, represent more biologically relevant conditions for drug discovery compared to standard growth media or buffers. These more complex environments may better reveal compounds with novel mechanisms of action that target bacterial pathogenicity factors essential for the integrity of non-tuberculous mycobacterial biofilms. Therefore, in this study, we utilized artificial sputum media and performed high-throughput screening to identify effective antibiofilm compounds in mycobacteria. We aimed to establish compounds with activity to inhibit biofilm formation as well as compromise established biofilms with an intention to use active compounds in combination with current anti-NTM antibiotics and improve treatment outcomes for biofilm infections.

## 2. Results

### 2.1. Optimization of High-Throughput Screening (HTS) Assay

The HTS assay was optimized using a 384-well plate (384WP) format in 24 technical replicates and a 96-well plate (96WP) format in 12 technical replicates by varying inoculum concentrations (Figure 1). The 384WP format was utilized for primary screening to identify active compounds, while the 96WP format was employed for secondary validation. This optimization aimed to determine the bacterial concentration, time points, and control antibiotic concentrations that promoted optimal mycobacterial biofilm formation and effectively assessed biofilm inhibition. MAB biofilms were generated under biologically relevant conditions using SCFM and evaluated over the course of a week using CV staining (Figure 1A). Furthermore, the antibiotic treatment groups alongside the DMSO control group were tested to identify parameters that produced the most significant differences in biofilm inhibition by day five, as measured by the percentage of inhibition between DMSO- and antibiotic-treated control wells (Figure 1B). Optimal results were achieved with biofilms containing 10^7^ CFU/mL followed by a five-day clarithromycin (CLA) treatment at a 70 μg/mL concentration. The MAB biofilms formed from 10^6^ CFU/mL inoculums were weak, resulting in high variability in CV staining and antibiotic treatment outcomes. In contrast, biofilms formed from 10^8^ CFU/mL inoculums showed minimal susceptibility to antibiotics (Figure 1B). Under selected conditions, the MAB biofilms and viability were assessed in the 96WP format. As shown in Figure 1C,D, a consistent and significant difference was observed between the bacterial growth control group (DMSO-treated wells) and the negative control group (CLA treatment). Optimal results were accomplished in a 96WP with biofilms generated using a 10^7^ CFU/mL inoculum and subjected to CLA treatment at a concentration of 70 μg/mL for five days. These parameters were selected later for the secondary confirmational assay.

### 2.2. Identification of Active Compounds Against Mycobacteria Biofilms

In the primary screen, a total of 24,000 compounds were tested, including 14,400 from the Maybridge compound library, 9000 from the ChemBridge DIVERSet compound library, and 600 from the Spectrum compound library, all sourced from the OSU/College of Pharmacy High-Throughput Screening Services Laboratory. MAB inoculum (10^7^ CFU/mL) was added to a 384WP preloaded with compounds at 10 μM in SCFM and, after staining the biofilm mass on day five, 354 compounds were identified as hits showing ≥ 30% biofilm inhibition. This corresponded to a primary screening hit rate of 1.48%. A secondary screening was conducted in a 96WP format to validate the hits. This stage identified 17 active compounds with ≥30% biofilm inhibition, yielding a hit rate of 4.8%. The selected hit compounds, listed in Table 1, were subsequently purchased from MolPort for the third confirmatory screening, where the biofilm-inhibitory activity was verified in three biological replicates, each performed with four technical replicates (Figure 2).

The antibacterial activity of 17 selected compounds was assessed against MAB under both pre- and post-biofilm formation conditions in SCFM (Figure 2A,B). Alongside this, bacterial viably was also measured using resazurin assay to determine whether biofilm inhibition resulted from bacterial killing or interference with the active mechanisms of biofilm formation. The results indicate that six compounds (**1**, **5**, **7**, **11**, **14**, and **16**) demonstrated high activity against MAB biofilms, achieving inhibition rates of over 60% and 40% in pre- and post-formation conditions, respectively. Of these six compounds, five also reduced bacterial viability by 40% or more under both conditions.

The quantification of bacterial biomass confirmed the effectiveness of select hit compounds in inhibiting biofilm formation in NTM clinical strains. We conducted additional investigations to evaluate the effectiveness of six selected compounds (**1**, **5**, **7**, **11**, **14**, and **16**) in preventing biofilm formation by MAB and MAH clinical isolates. As shown in Figure 3, all tested compounds significantly reduced biofilm formation relative to untreated controls. Notably, compounds **5** and **16** demonstrated the most substantial reduction in biofilm biomass across all tested isolates, highlighting their potential as effective agents against biofilm development in these NTM strains.

### 2.3. The Hit Compounds Exhibited Antimicrobial Activity in Replicating Mycobacteria In Vitro

To determine whether the compounds with antibiofilm activity also exhibited efficacy against replicating bacteria *in vitro*, we conducted antimicrobial susceptibility assays using mid-log-phase cultures of MAB19977 and MAH104. Two-fold serial dilutions of each compound were prepared in 1% DMSO and added to cultures grown in 7H9 broth. Bacterial viability was assessed by measuring the optical density at 600 nm (OD_600_) and by evaluating resazurin substrate oxidation rates on day 3 for MAB and day 4 for MAH. To further assess the efficacy of the hit compounds, we expanded this study to include clinical isolates of MAB and MAH obtained from CF patients. The MAB isolates included DNA01627, NR49093 (strain DJO44274), and NR44273 (strain 4529). For MAH, we tested the MAHA5 clinical isolate, a strain known for its robust biofilm formation and resilience to antimicrobial agents [20]. The 50% inhibitory concentrations of the compounds against these strains are summarized in Table 2. While the IC_50_ values varied between the two NTM species, the majority of the tested compounds demonstrated consistent *in vitro* activity across all tested strains.

In addition, we conducted bacterial survival studies using selected hit compounds that demonstrated *in vitro* activity against both MAB and MAH and exhibited antibiofilm properties. These compounds were tested in combination with the frontline antibiotic amikacin (AMK) to assess their potential to enhance antibiotic efficacy. The mid-log-phase cultures of MAB19977 and MAH104 were treated with either the compound alone at the corresponding IC_50_, AMK alone at two times the MIC, or a combination of the compound and AMK. The goal was to determine whether the hit compounds, potentially acting through novel mechanisms, could improve the effectiveness of AMK and enhance bacterial clearance compared to AMK alone. Our findings demonstrate the potential of selected hit compounds (**1**, **5**, and **16**) to exhibit synergistic effects with AMK, improving treatment efficacy against both mycobacterial strains (Figure 4).

### 2.4. Select Hit Compounds Demonstrated Potency Against Intracellular Mycobacteria Within Phagocytic Cells

To evaluate the potential of the six select hit compounds (**1**, **5**, **7**, **11**, **14**, and **16**) against intracellular mycobacteria, their cytotoxicity was assessed initially in differentiated THP-1 human macrophages. Using a resazurin oxidation assay and visual inspection of the cell monolayers, compounds were tested across a concentration range of 0.1 µM to 100 µM. The highest nontoxic concentrations for THP-1 cells were identified for each compound, as shown in Table 3. Subsequently, bacterial survival assays were performed at these nontoxic concentrations to determine the efficacy of the compounds against intracellular mycobacteria.

Among the six tested compounds, three demonstrated a significant reduction in the intracellular growth of MAB19977, measured as colony-forming units, while four compounds effectively decreased the viable MAH104 loads over a 5-day infection period (Figure 5). Compounds **1**, **5**, and **16** exhibited activities against both NTM strains, whereas compound **11** demonstrated potency in reducing MAH intracellular growth. It is important to note that certain compounds, when combined with bacterial infection, caused increased toxicity to THP-1 cells at previously identified nontoxic concentrations. These compounds are marked with asterisks in Table 3, and due to this toxicity, no data could be recorded for their efficacy in reducing intracellular bacterial loads.

## 3. Discussion

*Mycobacterium abscessus* causes one of the most challenging lung infections, particularly in patients with CF, due to its capacity to form resilient biofilms in the bronchial space [13]. These biofilms, also observed in the lung cavity of patients with COPD [36], are embedded in an extracellular matrix that enhances drug tolerance and results in poor treatment outcomes. Additionally, the dense cell aggregation within the biofilms, combined with the mucus-rich lung environment, restricts oxygen diffusion. This creates a hypoxic environment which, along with other lung-specific factors, promotes the metabolic remodeling of the pathogen, driving its transition into a persistent, non-replicative state and further exacerbating its resistance to antimicrobial therapy and complicating therapeutic interventions [17,37]. Current treatment options remain limited, with no standardized regimens or consistently effective therapies [5,38,39]. Multidrug regimens, often involving aminoglycosides, macrolides, and other antibiotics, offer partial relief but demonstrate poor success rates, particularly in the presence of resistance genes [40,41].

In this study, we employed artificial sputum media to simulate the cystic fibrosis lung environment and conducted high-throughput screening to identify compounds with antibiofilm activity against MAB. Our goal was to identify novel compounds under clinically relevant conditions with potent antibiofilm activity to enhance the efficacy of antibiotics against biofilm-associated bacteria. A total of 24,000 compounds were screened from the Maybridge, ChemBridge DIVERSet, and Spectrum compound libraries, resulting in the identification of 354 potential hits with ≥30% biofilm inhibition in the primary screen. After further validation in secondary and confirmatory screens, 17 active compounds were selected. Six compounds exhibited robust biofilm inhibition, with over 60% inhibition in pre-formation assays and over 40% in post-formation assays. Notably, five of these compounds also reduced bacterial viability by at least 40%, highlighting their dual action in preventing biofilm formation and reducing bacterial survival. The *in vitro* antimicrobial susceptibility tests showed that several of these compounds had consistent activity against both MAB and MAH clinical isolates, particularly compounds **1**, **5**, **11**, and **16**. Additionally, combining the hit compounds with AMK, one of the most important antibiotics in the aminoglycoside class for treating pulmonary NTM infections, enhanced bacterial clearance *in vitro*, highlighting potential synergistic effects. The further evaluation of select compounds for their ability to reduce intracellular mycobacterial loads in human macrophages revealed that compounds **1**, **5**, and **16** significantly decreased the intracellular survival of both MAB and MAH. We also observed that some compounds induced toxicity at nontoxic concentrations during infection, highlighting the need for the careful evaluation of their therapeutic potential. Overall, our findings suggest that certain antibiofilm compounds could be valuable adjuncts for improving current antibiotic effectiveness in treating MAB and other NTM infections.

In this study, we identified compounds belonging to various clusters, including pyrimidinone, pyrimidine, benzothiazole, sulfonylphenyl, phenothiazine, benzoxazole, pyrazolopyridine, quinazoline, thiourea, thiazole, and others. Some of these compound classes have been previously identified through the high-throughput screening of large chemical libraries and are known for their antimicrobial activities, including against *M. tuberculosis (Mtb)*. For instance, ethacridine (compound **1**), known for its broad-spectrum antimicrobial properties and effectiveness against various bacteria [42], including *Mtb* [43,44], and within biofilms of Staphylococcus aureus [45], was found in our study to effectively disrupt biofilm formation in MAB. This FDA-approved drug primarily targets the MtopI enzyme, which is essential for DNA replication, transcription, recombination, and chromosome condensation in Mtb, making it a promising candidate for disrupting bacterial viability. Also, its ability to enhance the activity of moxifloxacin against Mtb suggests a potential synergistic effect, offering promise for treating multidrug-resistant tuberculosis (MDR-TB) [46]. With its mechanism of action, involving interference with DNA topology, ethacridine could be a valuable compound for both preventing and treating biofilm-related non-tuberculous mycobacteria infections.

Compounds **5** and **16** exhibited significant antibacterial activity by not only inhibiting bacterial viability but also demonstrating potent efficacy against pre-formed biofilms. Additionally, these compounds maintained their activity under diverse conditions, including *in vitro* replicative environments and intracellular states. This suggests that the compounds target a critical mechanism in NTM that is universally expressed and indispensable across all stages of bacterial growth and survival, making them promising candidates for broad-spectrum therapeutic applications. Compound **5** belongs to the phenothiazine group which has been recognized for its ability to inhibit multidrug efflux pump activity in *S. aureus* [47] and type IV pilus production in multiple Gram-negative bacterial species [48], as well as for being active against *Enterococcus faecalis* and *Enterococcus faecium* human and animal clinical isolates [49] while effectively reducing biofilm mass in *Enterococcus faecalis* [50]. Phenothiazines have also demonstrated both *in vitro* and *in vivo* activity against Mtb, including MDR-TB [51]. These compounds disrupt biofilms by targeting the quorum-sensing systems of sessile bacterial populations [52], which are often more resistant to antibiotics, thereby enhancing overall antimicrobial efficacy. The synergistic interaction between phenothiazines and other antimicrobial agents further suggests their potential as therapeutic agents for chronic infections [53,54].

Furthermore, Compound **16**, a derivative of fluoren, a tricyclic aromatic hydrocarbon commonly found in various pharmaceutical compounds, was found in our study to exhibit high *in vitro*, antibiofilm, and intracellular activities against MAB. Numerous fluorene derivatives have demonstrated broad-spectrum antimicrobial activity against a wide range of microorganisms, including both Gram-positive and Gram-negative bacterial strains as well as yeasts, with some outperforming standard antibiotics [55]. Additionally, research highlights that certain fluorene-based compounds exhibit significant activity against mycobacteria, particularly in Mtb and *M. bovis*, including drug-resistant strains, showcasing their potential as novel anti-tuberculosis drugs [56]. Studies have explored structural modifications of fluorene with diverse functional groups to enhance antimycobacterial efficacy and elucidate their mechanisms of action. Notably, some derivatives show promising activity against both replicating and non-replicating bacterial populations, including intracellular Mtb within infected macrophages [57,58,59]. These broad-spectrum antimicrobial properties and the ability to inhibit biofilm formation underscore their potential for further investigation in the treatment of infections caused by NTM.

Compound **2**, N-benzyl-N-methyl-6-phenylpyrimidin-4-amine, is a pyrimidine derivative, a class of compounds frequently investigated for their ability to inhibit enzymes involved in DNA synthesis [60,61]. Given its chemical structure and mechanism of action, this compound aligns with other pyrimidine analogs that have been studied for their potential efficacy against antibiotic-resistant strains and biofilm formation, which are major challenges in treating persistent infections [62,63]. Compound **3**, N-(1-benzothiophen-3-yl)-N′-(4-fluorophenyl)urea, which belongs to benzothiazole cluster, demonstrated notable potential for MAB biofilm disruption. Benzothiazole-based compounds have shown promising antibacterial activity against a variety of bacteria, including Gram-positive and Gram-negative bacteria, and Mtb. They can bind to different targets in bacterial cells, such as enzymes involved in cell-wall synthesis, cell division, and DNA replication [64,65]. Interestingly, benzothiazole–urea hybrids have demonstrated antibacterial and anti-biofilm activity against *S. aureus*, including MRSA being almost as effective as vancomycin in reducing bacterial load in a mouse model of abdominal infection [66]. Other compounds in this study, including sulfonylphenyl and benzoxazole derivatives, are not widely reported for their anti-mycobacterial or antibiofilm activities. However, based on their known ability to interact with bacterial membranes and metabolic pathways, they may possess properties that suggest potential for disrupting biofilm integrity. Compounds **13** and **14**, which are fluorescent dyes, contrast with compound **17**, a thiazole-based compound identified as 2-(4-(4-Chlorophenyl)-1,3-thiazol-2-yl)isoindoline-1,3-dione. It is well documented that thiazole, a dominant pharmacophore widely present in antibacterial drugs, can exhibit significant antibacterial activity. For instance, novel thiazole nortopsentin analogs were recently synthesized and evaluated for their ability to inhibit biofilm formation in clinically relevant Gram-positive and Gram-negative pathogens. These compounds were found to interfere with the initial step of biofilm formation in a dose-dependent manner, showing selective activity against staphylococcal strains [67].

## 4. Materials and Methods

### 4.1. Bacterial Strains and Growth Culture

*Mycobacterium abscessus* subsp. *abscessus* (MAB) strain 19977 was obtained from the American Type Culture Collection (ATCC). *Mycobacterium avium* subsp. *hominissuis* 104 (MAH104) and *Mycobacterium avium* subsp. *hominissuis* A5 (MAHA5) were isolated from the blood of a patient with AIDS [68]. Clinical isolates of MAB, including DNA01627, NR49093 strain DJO44274, and NR44273 strain 4529, were provided in collaboration with the Cystic Fibrosis Research and Development Program at National Jewish Health, Denver, CO, USA. All studies were conducted in compliance with Oregon State University Institutional Biosafety Committee approval, protocol #3623. Bacteria were cultured on Middlebrook 7H10 agar or in Middlebrook 7H9 broth (Difco Laboratories, Detroit, MI, USA) supplemented with 10% oleic acid, albumin, dextrose, and catalase (OADC; Hardy Diagnostics, Santa Maria, CA, USA), along with glycerol. To prepare bacterial suspensions, mid-log-phase cultures grown on 7H10 agar plates were resuspended in Hanks’ Balanced Salt Solution (HBSS; VWR, Visalia, CA, USA) and adjusted to a McFarland standard of 3.0 (~9 × 10^8^ cells/mL). The suspensions were sonicated for 1 min in a Branson ultrasonic bath and passed through a 22-gauge syringe 10 times to minimize clumping. Following this, the inoculums were allowed to settle for 15 min, and the top portion of the suspension was transferred to a new tube. The optical density at 600 nm (OD_600_) was measured (1.0 OD_600_~3 × 10^8^ cells/mL), and the suspensions were further diluted in SCFM or RPMI-1640 mammalian cell culture media to the desired concentrations. Additionally, the sonicated inoculums were serially diluted and plated onto 7H10 agar plates to ensure precise inoculum concentrations.

### 4.2. Synthetic Cystic Fibrosis Sputum Media

SCFM was formulated based on the average concentrations of ions, free amino acids, glucose, and lactate observed in the CF sputum samples [69] and is outlined in Table 4. Amino acids were prepared as 100 mM stock solutions in deionized water and stored in the dark at 4 °C. Specific amino acids, such as tyrosine, aspartate, and tryptophan, were dissolved in 1.0 M, 0.5 M, and 0.2 M NaOH, respectively, to enhance solubility. Lactate stock solutions were adjusted to a pH of 7.0 using NaOH. All components were combined sequentially, and the final solution was adjusted to a pH of 6.8. The medium was then sterilized by filtration using 0.2 μm pore-size filtration flasks (Corning, Glendale, CA, USA) and stored at 4 °C until use.

### 4.3. Chemicals and Compound Libraries

The screening process involved a total of 24,000 compounds, sourced from three distinct libraries. Of these, 14,400 compounds were from the Maybridge compound library, 9000 from the Cambridge compound library, and 600 from the Spectrum compound library. All compounds were obtained from the Oregon State University/College of Pharmacy High-Throughput Screening Services Laboratory. The selection of compounds was designed to maximize chemical diversity, thereby increasing the likelihood of identifying biologically active candidates. The compounds were supplied in two formats: (1) as a single-dose preparation at a concentration of 10 μM in SCFM dispensed in a 384WP with a total volume of 40 μL per well and (2) as a stock solution at a concentration of 100 μM in 1% DMSO used as the solvent carrier, provided in a 96WP.

Antibiotics included amikacin (AMK), purchased from Research Products International (Mt. Prospect, IL, USA), and clarithromycin (CLA), obtained from Tokyo Chemical Industry (Tokyo, Japan). Crystal violet, used for biofilm staining, was purchased from Sigma-Aldrich. Additional reagents included Phorbol 12-myristate 13-acetate (PMA) and resazurin, both sourced from Sigma-Aldrich (St. Louis, MO, USA). All compounds and reagents were prepared and stored according to the manufacturers’ specifications to ensure optimal stability and consistency throughout the experiment.

### 4.4. Compound Screening

Ten microliters of MAB inoculum, prepared at a concentration of 9 × 10^8^ CFU/mL in SCFM, were added to each well of a 384WP pre-loaded with compounds in 40 μL of SCFM at a final concentration of 10 μM. The plates were incubated at 37 °C for up to five days. As a control for 100% inhibition, the last two columns of wells received 70 μg/mL CLA. Additionally, DMSO was included as a control to assess any potential effects on bacterial growth and/or biofilm formation.

Compounds that inhibited bacterial biofilm formation by more than 30%, assessed by crystal violet staining, described below, were identified as potential hits. These compounds were subsequently cherry-picked from the stock library of 100 μM in 1% DMSO and subjected to a secondary assay in 96-well round-bottom plates, performed in triplicate under the same conditions as the primary screen. A total of 100 μL of bacterial inoculum at a concentration of 3 × 10^8^ CFU/mL in SCFM was dispensed into each well of a 96WP. Compounds were added to each well at a final concentration of 10 μM, and the plates were incubated at 37 °C for up to five days. Compounds that demonstrated consistent activity in both the primary and secondary assays were purchased from MolPort and further validated as hit compounds through three independent biological replicates, each performed in triplicate using a 96WP format.

### 4.5. Bacterial Viability Using Resazurin Staining

Resazurin staining was conducted prior to biofilm visualization only in the secondary and confirmation assay with purchased compounds. These assays focused on selected hit compounds, incorporating both bacterial growth and killing controls to ensure accurate evaluation. Briefly, the planktonic solution was gently removed to preserve biofilm integrity. The biofilms were washed once with phosphate-buffered saline (PBS) and incubated with resazurin at 5 μg/well, protected from light, at room temperature for up to 2–3 h or until the appropriate color developed in the bacterial growth control (without compound exposure). Fluorescence was measured at an excitation/emission wavelength of 530 nm/590 nm using a Tecan Infinite F200 Fluorescent Microplate Reader. In addition, compound activity was assessed in both planktonic and biofilm-associated bacteria by serially diluting the supernatants and biofilm bacteria, followed by the quantification of colony-forming units on 7H10 agar plates. This assay was conducted exclusively with the hit compounds.

### 4.6. Crystal Violet (CV) Staining

To measure the biomass, briefly, the resazurin solution was gently removed from the plates. The wells were fixed with 95% ethanol (40 μL for 384WP and 200 μL for 96WP) for 10 min at room temperature and then air-dried. Next, the wells were stained with 0.1% crystal violet (Sigma-Aldrich, St. Louis, MO, USA) for 10 min. After staining, the wells were washed twice with deionized water and allowed to air-dry for 10 min. To quantify biofilm formation, the stained biofilms were solubilized in 30% acetic acid for 30 min and transferred to 384- or 96-well optically clear flat-bottom plates. The absorbance was measured using an Epoch Microplate Spectrophotometer (BioTek, Winooski, VT, USA) at a wavelength of 570 nm.

The percentage of biofilm inhibition was calculated using the following formula:$$\%\;inhibition\;=\;100\;\times \;[1-\frac{\left(Drug\;treated\;well\;average\right)\;-\;\left(Negative\;control\;well\;average\right)}{\left(Positive\;control\;well\;average\right)-\left(Negative\;control\;well\;average\right)}]$$
where the drug-treated well average is the absorbance value from wells treated with the test compound. The positive control well average is the absorbance value from wells containing only the bacterial culture and no compound. The negative control well average is the absorbance value from wells treated with the antibiotic CLA70.

### 4.7. Compound Activity in Pre- and Post-Biofilm Formation

The activity of the compounds was additionally assessed against clinical isolates of MAB and MAH in two contexts: the inhibition of biofilm formation (pre-biofilm condition) and disruption of formed biofilms (post-biofilm condition). For the inhibition of biofilms, the assay was carried out with the selected hit compounds, as detailed above. Biofilm formation was quantified using CV staining, while bacterial viability was assessed through the redox indicator resazurin dye, with assessments conducted for both assays on day 5 following compound addition.

The activity of the selected hit compound was evaluated with post-formed biofilms under the following conditions: A total of 100 μL of bacterial inoculum at a concentration of 3 × 10^8^ CFU/mL in SCFM was dispensed into each well of a 96WP and incubated at 37 °C for three days to allow biofilm formation. Subsequently, the compound was added to each well at a final concentration of 10 μM. The bacterial viability and biofilm mass were then evaluated on day 5 using resazurin and CV staining, respectively, as detailed above.

### 4.8. Assay for 50% Inhibitory Concentration (IC50)

To establish the minimum inhibitory concentration (MIC) required to inhibit 50% of the pathogen, bacterial inoculums of MAB19977, MAH104, or clinical isolates were prepared in 7H9 media supplemented with OADC, 10% glycerol, and 0.1% Tween 80. The bacterial suspension was adjusted to an OD_600_ of 0.5 and diluted to 0.03–0.05 in 7H9 media prior to compound addition. Compounds were prepared in DMSO at concentrations ranging from 10 μM to 0.01 μM by half-log serial dilutions. CLA at 70 μg/mL and 1% DMSO served as positive and negative controls, respectively. The plates were incubated statically for 72 h for MAB19977 and 96 h for MAH104. Resazurin was then added to the plates at a final concentration of 5 μg/well and incubated at 37 °C until a color change was observed. Fluorescence was measured at 530 nm/590 nm using a Tecan plate reader, and the percentage of inhibition was calculated using the formula described above. The experiment was conducted in two biological replicates.

### 4.9. Assay for 50% Cytotoxic Concentration (CC50)

Human-derived THP-1 monocytes (ATCC TIB-202) were cultured in RPMI-1640 medium (Cellgro, Manassas, VA, USA) supplemented with 10% heat-inactivated fetal bovine serum, 2 mM L-glutamine, and 25 mM HEPES and maintained at 37 °C with 5% CO_2_. To differentiate the THP-1 cells into macrophages, they were plated at 1 × 10^5^ cells per well in a clear, flat-bottom 96WP with culture media and treated overnight with 30 ng/mL of phorbol-12-myristate-13-acetate (PMA). The following day, the wells were replenished with fresh media, and the cells were allowed to rest for an additional two days at 37 °C with 5% CO_2._ The hit compounds were serially diluted in DMSO, ranging from 10 μM to 0.01 μM, using half-log dilutions. To determine the cytotoxic concentration inducing 50% cell death (IC_50_), 4 μL of each compound dilution was added to 196 μL of RPMI-1640 medium and dispensed onto the cell monolayers in each well of a 96WP. Cyclosporine A (50 μg/mL) was used as a positive control for cytotoxicity, while 1% DMSO served as the negative control. After incubating the plates at 37 °C with 5% CO_2_ for 72 h, cytotoxicity was assessed using the resazurin assay by adding 10 μL of 50 μg/mL resazurin in PBS to each well. The plates were incubated for 2–3 h, and fluorescence was measured at 530 nm/590 nm using a Tecan plate reader. Additionally, cell monolayers were visually examined under an inverted microscope to detect any signs of cytotoxicity. The cytotoxic concentration was identified as the lowest compound concentration that neither caused a color change in resazurin nor induced visible damage to the cell monolayer. This concentration was used to calculate the IC_50_, representing the concentration at which 50% of the host cells were killed.

### 4.10. Compound Activity in Macrophages

The THP-1 monolayers were established in a 96WP, as previously described, and bacterial infections were conducted with a multiplicity of infection (MOI) of 1 for MAB19977 and a MOI of 5 for MAH104. After a 2 h infection period, cells were washed with PBS and treated with 200 μg/mL AMK for 2 h to kill any extracellular bacteria. The wells were then replenished with fresh media containing the tested compounds at their highest nontoxic concentrations and, subsequently, every other day throughout the duration of the experiment. Additionally, antibiotics were added at bactericidal concentrations of 32 μg/mL AMK and 4 μg/mL CLA for both pathogens. After 5 days of incubation at 37 °C with 5% CO_2_, monolayers were lysed 0.1% Triton-X-100 and viable bacteria were enumerated by plating serial dilutions on 7H10 agar plates. Data were calculated as the percentage of surviving bacteria relative to the DMSO-only control.

### 4.11. Statistical Analysis

Statistical analyses were performed using Student’s *t*-test for comparisons between two groups and one-way analysis of variance (ANOVA) for comparisons across multiple groups, followed by post hoc multiple-comparison tests to assess pairwise differences between the DMSO control and experimental treatment groups. Results were derived from three biological replicates, each performed in three technical replicates unless otherwise stated. Statistical analyses and graphical visualizations were carried out using GraphPad Prism (version 10.0). A *p*-value of less than 0.05 was considered statistically significant (* *p* < 0.05; ** *p* < 0.01; *** *p* < 0.001), with specific values reported in figure legends and indicated by asterisks in graphical outputs. All data are presented as the mean ± standard deviation (SD), with error bars included in graphical representations to reflect variability within replicates.

## 5. Conclusions

This study aimed to identify compounds with anti-NTM biofilm activity through the high-throughput screening of a diverse compound library under conditions simulating a CF patient’s sputum. Our findings revealed several promising candidates with varying degrees of biofilm inhibition and antimicrobial activity against clinical NTM strains. Notably, compounds such as ethacridine, phenothiazine, and fluorene derivatives demonstrated significant potential by not only preventing biofilm formation but also enhancing antimicrobial efficacy and markedly reducing intracellular MAB loads in macrophages. These results suggest that the identified compounds likely target a critical, universally expressed mechanism in NTM, essential for bacterial growth and survival across different stages. This positions them as promising candidates for broad-spectrum therapeutic applications. However, the precise molecular mechanisms underlying their activity remain unclear and require further elucidation through molecular and biochemical studies. While these findings highlight their potential as valuable candidates for further development as adjuncts to existing antibiotics, it is challenging to assess their draggability at this stage without *in vivo* studies. Also, some chemical modifications may be necessary to optimize their activity and minimize toxicity for future therapeutic use. Notably, certain compounds, as highlighted in this paper, exhibited increased toxicity when tested in the phagocytic cell infection model. These findings underscore the importance of *in vivo* studies to better understand the clinical relevance and safety of these compounds.

## Figures and Tables

**Figure 1 pharmaceuticals-18-00225-f001:**
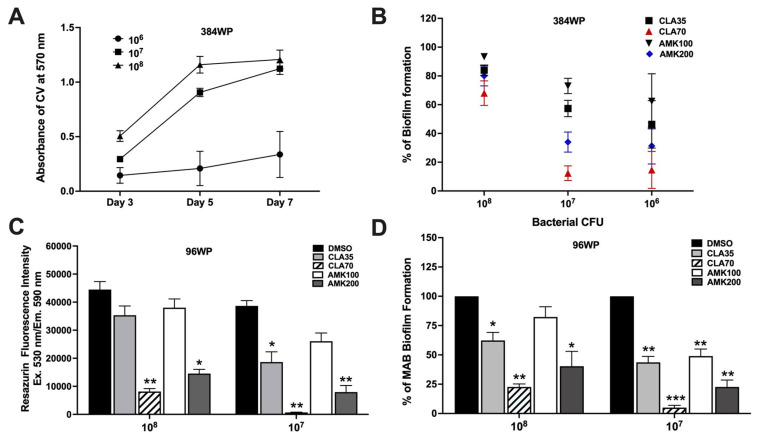
The development of the HTS assay for the identification of compounds with potency against mycobacterial biofilms. (**A**) The CV absorbance readings were recorded for MAB grown in SCFM over time and across a range of concentrations in the 384WP format. (**B**) The percentage of biofilm formation on day five was calculated as described in Section 4 using DMSO- and antibiotic-treated wells as controls for bacterial growth and inhibition in the 384WP format. (**C**) MAB viability was recorded as fluorescent readings for the control and antibiotic-treated groups, based on resazurin reduction rates in the 96WP format. (**D**) The percentage of MAB biofilm formation on day 5 was determined based on the CV assay. Data are expressed as the means ± standard deviations (SD) of three independent experiments. The statistical significance between the DMSO and antibiotic-treated groups is indicated as follows: * *p* < 0.05; ** *p* < 0.01; and *** *p* < 0.001.

**Figure 2 pharmaceuticals-18-00225-f002:**
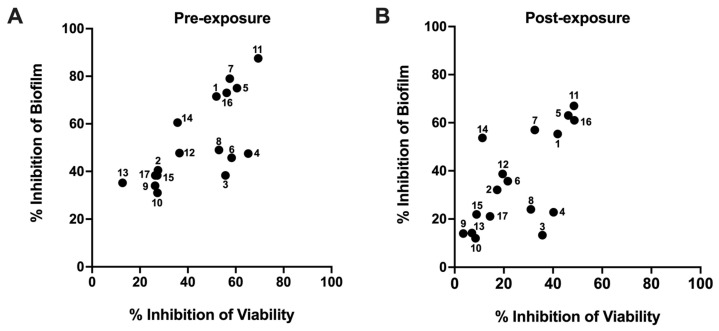
Activity of selected hit compounds against biofilms of MAB19977 under pre- and post-formation conditions. (**A**) Prevention of biofilm formation (pre-exposure condition). (**B**) Inhibition of 24 h pre-formed biofilms (post-exposure condition). Data are presented as percentage of inhibition in bacterial viability in biofilms or biomass relative to untreated control biofilms, based on three independent biological replicates.

**Figure 3 pharmaceuticals-18-00225-f003:**
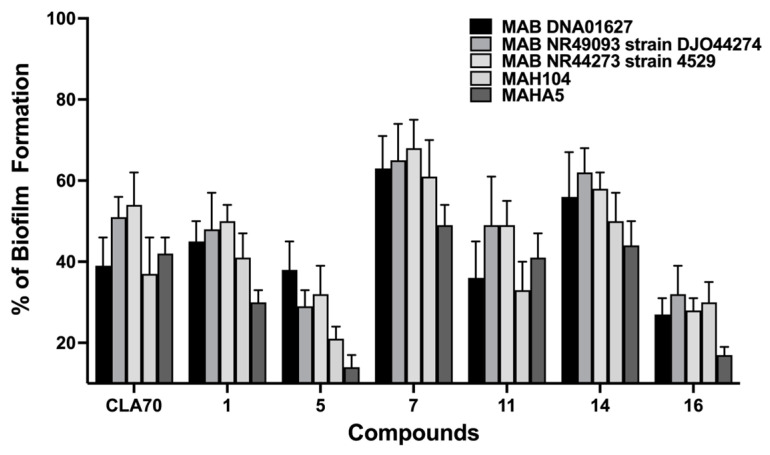
The screening of antibiofilm activity across NTM strains. Biofilm biomass was quantified using crystal violet staining under pre-treatment conditions with select hit compounds at a 100 μM concentration, as described in the Materials and Methods. DMSO and CLA70 served as growth and inhibition controls, respectively. Data are presented as the mean ± standard deviation from three independent replicates.

**Figure 4 pharmaceuticals-18-00225-f004:**
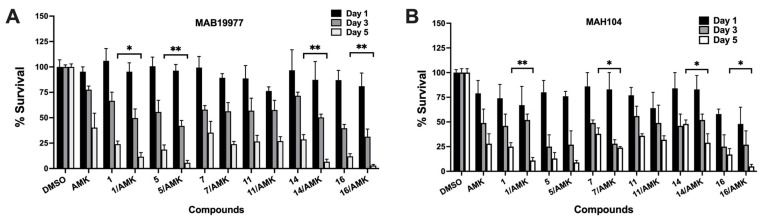
The in vitro time-kill dynamics of mycobacteria during treatment with select hit compounds and in combination with an antibiotic. (**A**) MAB19977 survival rates in AMK, compound, and compound–AMK treatment groups over 5 days in 7H9 broth. (**B**) MAH104 survival rates in AMK, compound, and compound–AMK combination treatment groups over 5 days in 7H9 broth. Bacterial CFUs were recorded after treatment with the antibiotic at 2× the MIC and/or compounds at IC_50_ concentrations. Survival percentages were calculated relative to the DMSO growth control (untreated). Antimicrobials were added at time zero and then supplemented every other day throughout the duration of the experiment. Data are presented as means ± standard deviations (SD) from two independent experiments performed in triplicate. The statistical significance between the compound group alone and antibiotic–compound combination treatment groups on day 5 is indicated as * *p* < 0.05 and ** *p* < 0.01.

**Figure 5 pharmaceuticals-18-00225-f005:**
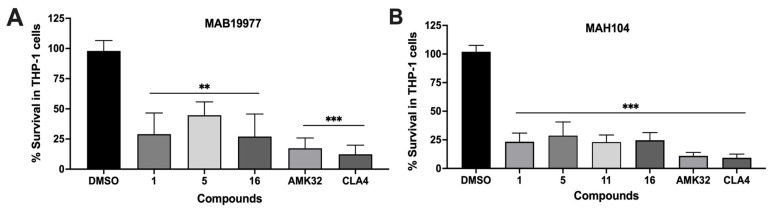
The quantification of bacterial CFUs for establishing the intracellular potency of hit compounds in infected THP-1 macrophages. Compounds at their highest nontoxic concentrations (as listed in Table 3) were added to (**A**) MAB19977-infected or (**B**) MAH104-infected THP-1 cell monolayers at 2 h post-infection and, subsequently, every other day throughout the duration of the experiment. Bacterial CFUs were determined by lysing the cells with 0.1% Triton X-100 on day 5 for both MAB19977 and MAH104, followed by plating serial dilutions on 7H10 agar plates. The percentage of surviving bacteria was calculated by dividing the CFUs per well for each treatment group by the CFUs in the DMSO growth control, which represented a 100% bacterial survival. AMK and CLA at a bactericidal concentration of 32 µg/mL and 4 µg/mL, respectively, were used as controls to assess the growth inhibition of MAB19977 and MAH104 within macrophages. The tested compounds demonstrated a statistically significant reduction in intracellular bacterial survival as determined by one-way ANOVA with multiple comparisons between the DMSO control and compound-treated groups. The significance between the DMSO control group and compound treatment groups on day 5 is indicated as ** *p* < 0.01 and *** *p* < 0.001.

**Table 1 pharmaceuticals-18-00225-t001:** The list of active compounds identified in this study.

Compound	Structure	SMILES Molecular Formula	Name
**1**	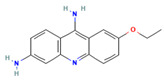	CCOc1ccc2nc3cc(ccc3c(c2c1)N)N C_15_H_15_N_3_O	Ethacridine
**2**	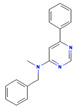	CN(Cc1ccccc1)c1cc(ncn1)-c1ccccc1 C_18_H_17_N_3_	N-benzyl-N-methyl-6-phenylpyrimidin-4-amine
**3**	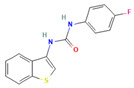	Fc1ccc(cc1)NC(=O)Nc1csc2ccccc12 C_15_H_11_FN_2_OS	N-(1-benzothiophen-3-yl)-N′-(4-fluorophenyl)urea
**4**	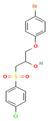	OC(COc1ccc(cc1)Br)CS(=O)(=O)c1ccc(cc1)Cl C_15_H_14_BrClO_4_S	1-(4-Bromophenoxy)-3-(4-chlorophenyl)sulfonylpropan-2-ol
**5**	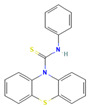	S=C(Nc1ccccc1)N1c2ccccc2Sc2ccccc21 C_19_H_14_N_2_S_2_	N-phenylphenothiazine-10-carbothioamide
**6**	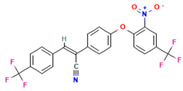	[O-][N+](=O)c1cc(ccc1Oc1ccc(cc1)/C(=C/c1ccc(cc1)C(F)(F)F)C#N)C(F)(F)F C_23_H_12_F_6_N_2_O_3_	(Z)-2-[4-[2-nitro-4-(trifluoromethyl)phenoxy]phenyl]-3-[4-(trifluoromethyl)phenyl]prop-2-enenitrile
**7**	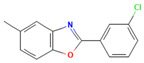	Cc1ccc2oc(nc2c1)-c1cccc(c1)Cl C_14_H_10_ClNO	2-(3-Chlorophenyl)-5-methylbenzo[d]oxazole
**8**	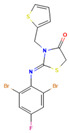	Fc1cc(c(c(c1)Br)/N=C/1\SCC(=O)N1Cc1cccs1)Br C_14_H_9_Br_2_FN_2_OS_2_	2-(2,6-Dibromo-4-fluorophenyl)imino-3-(thiophen-2-ylmethyl)-1,3-thiazolidin-4-one
**9**	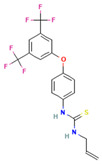	FC(F)(F)c1cc(cc(c1)C(F)(F)F)Oc1ccc(cc1)NC(=S)NCC=C C_18_H_14_F_6_N_2_OS	1-[4-[3,5-Bis(trifluoromethyl)phenoxy]phenyl]-3-prop-2-enylthiourea
**10**	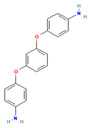	Nc1ccc(cc1)Oc1cccc(c1)Oc1ccc(cc1)N C_18_H_16_N_2_O_2_	1,3-Bis(4-aminophenoxy)benzene
**11**	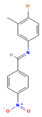	Cc1cc(ccc1Br)/N=C/c1ccc(cc1)[N+](=O)[O-] C_14_H_11_BrN_2_O_2_	N-(4-bromo-3-methylphenyl)-1-(4-nitrophenyl)methanimine
**12**	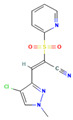	Cn1cc(c(n1)/C=C(\C#N)S(=O)(=O)c1ccccn1)Cl C_12_H_9_ClN_4_O_2_S	(E)-3-(4-chloro-1-methylpyrazol-3-yl)-2-pyridin-2-ylsulfonylprop-2-enenitrile
**13**	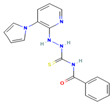	O=C(NC(=S)NNc1ncccc1-n1cccc1)c1ccccc1 C_17_H_15_N_5_OS	Cyto10B8
**14**	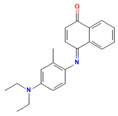	CCN(CC)c1ccc(c(c1)C)/N=C\1/C=CC(=O)c2ccccc21 C_21_H_22_N_2_O	4-((4-(Diethylamino)-2-methylphenyl)imino)naphthalen-1(4H)-one
**15**	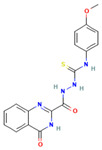	COc1ccc(cc1)NC(=S)NNC(=O)C1=Nc2ccccc2C(=O)N1 C_17_H_15_N_5_O_3_S	1-(4-methoxyphenyl)-3-[(4-oxo-3H-quinazoline-2-carbonyl)amino]thiourea
**16**	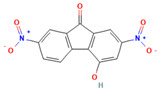	Oc1cc(cc2c1-c1ccc(cc1C2=O)[N+](=O)[O-])[N+](=O)[O-] C_13_H_6_N_2_O_6_	4-Hydroxy-2,7-dinitrofluoren-9-one
**17**	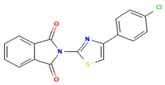	Clc1ccc(cc1)-c1csc(n1)N1C(=O)c2ccccc2C1=O C_17_H_9_ClN_2_O_2_S	2-(4-(4-Chlorophenyl)thiazol-2-yl)isoindoline-1,3-dione

**Table 2 pharmaceuticals-18-00225-t002:** *In vitro* activity of hit compounds against NTM strains.

Compound	IC_50_ [μM]					
	MAB19977	DNA 01627	NR49093 Strain DJO44274	NR44273 Strain 4529	MAH104	MAHA5
**1**	3	10	10	3	10	10
**2**	10	10	10	10	10	10
**3**	32	32	32	32	100	100
**4**	100	100	32	32	-	-
**5**	32	32	32	32	10	10
**6**	3	3	10	10	3	3
**7**	100	100	100	100	32	32
**8**	3	3	3	3	10	10
**9**	10	-	-	-	32	100
**10**	10	10	32	32	-	-
**11**	10	10	32	32	100	100
**12**	10	10	-	-	100	100
**13**	32	32	10	32	10	10
**14**	10	10	10	10	10	10
**15**	100	-	-	-	32	100
**16**	3	3	3	3	10	10
**17**	10	-	-	-	10	-

**Table 3 pharmaceuticals-18-00225-t003:** Activity of select hit compounds in THP-1 macrophages.

Compound	THP-1 Cytotoxicity [μM]	Intracellular Killing in THP-1 Cells	
		MAB19977	MAH104
**1**	32	Yes	Yes
**5**	100	Yes	Yes
**7**	100	*	*
**11**	32	*	Yes
**14**	10	*	*
**16**	32	Yes	Yes

Note: for active compounds marked with an asterisk, bacterial infection increased the toxicity to THP-1 cells at concentrations that were initially nontoxic.

**Table 4 pharmaceuticals-18-00225-t004:** Synthetic cystic fibrosis media ingredient list.

Order	Reagent Name	Stock Solution	Volume for 1 L
1	NaH_2_PO_4_	0.2 M	6.5 mL
2	Na_2_HPO_4_	0.2 M	6.25 mL
3	KNO_3_	1 M	0.348 mL
4	NH_4_Cl	-	0.122 g
5	KCl	-	1.114 g
6	NaCl	-	3.03 g
7	MOPS	10 mM	10 mL
8	DI H2O	-	779.6 mL
9	l-aspartate,	0.5 M in NaOH	8.27 mL
10	l-threonine	100mM	10.72 mL
11	l-serine	100 mM	14.46 mL
12	l-glutamate·HCl	100 mM	15.49 mL
13	l-proline	100 mM	16.61 mL
14	l-glycine	100 mM	12.03 mL
15	l-alanine	100 mM	17.8 mL
16	l-cysteine·HCl	100 mM	1.6 mL
17	l-valine	100 mM	11.17 mL
18	l-methionine	100 mM	6.33 mL
19	l-isoleucine	100 mM	11.2 mL
20	l-leucine	100 mM	16.09 mL
21	l-tyrosine	1 M in NaOH	8.02 mL
22	l-phenylalanine	100 mM	5.3 mL
23	l-ornithine·HCl,	100 mM	6.76 mL
24	l-lysine·HCl,	100 mM	21.28 mL
25	l-histidine·HCl,	100 mM	5.19 mL
26	l-tryptophan	0.2 M in NaOH	0.13 mL
27	l-arginine·HCl	100 mM	3.06 mL
28	CaCl_2_	1 M	1.754 mL
29	MgCl_2_	1 M	0.606 mL
30	FeSO_4_·7H_2_O	3.6 mM	1 mL
31	d-glucose	1 M	3 mL
32	l-lactate	1 M	9.3 mL

## Data Availability

The original contributions presented in this study are included in the article. Further inquiries can be directed to the corresponding author.

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
