# Peer review of "Discovery of Biofilm-Inhibiting Compounds to Enhance Antibiotic Effectiveness Against M. abscessus Infections"

_pharmaceuticals, 2025, doi:10.3390/ph18020225_

Round 1
Reviewer 1 Report
Comments and Suggestions for Authors
In the manuscript titled "Discovery of Biofilm-Inhibiting Compounds to Enhance Antibiotic Effectiveness Against Mycobacterial Infections," the authors have screened a library of small molecules to identify compounds that inhibit biofilm formation in selected Mycobacterium species. This effort to screen a large compound library for potential biofilm-inhibiting agents is commendable. The study reports the identification of 17 active biofilm-inhibiting compounds from the screening, presenting promising results. However, I have several queries and suggestions for the authors to address:
-
Line 55 and 381: Does SCFM stand for Synthetic Cystic Fibrosis Medium or Cystic Fibrosis Sputum Medium? Please clarify.
-
Line 91: A reference for bacterial amyloids should be added.
-
Statistical analyses and p-values are missing from the figures and methods sections. Please include these to strengthen the study's validity.
-
Line 169: Compounds 5 and 16 demonstrate significant biofilm inhibition but also exhibit high viability inhibition. How do the authors justify these findings?
-
Is AMK referring to an aminoglycoside class of antibiotics or specifically to Amikacin? This ambiguity leads to confusion while reading the manuscript.
-
Figure 5: In addition to the percentage graph, providing a CFU graph (log/CFU) would help readers better understand the extent of log reduction achieved after treatment.
-
Line 376: What conditions were used for sonication? Improper sonication can lead to bacterial killing or cell wall breakage, which might affect the results.
-
For Synthetic Cystic Fibrosis Medium, was the media composition formulated by the authors or adapted from previous publications? Please specify.
-
Line 453: The sentence is incomplete. A thorough proofreading of the manuscript is recommended before resubmission.
-
Line 454: Post-biofilm methods are mentioned in the heading but not described in the methods section.
-
Please provide the full forms of IC50 and CC50 for clarity.
-
How do the compounds penetrate the cells to kill intracellular bacteria, while Amikacin does not?
-
Including CV staining images in the figures would make the manuscript more engaging for readers.
- The authors have used clinical strains in this study. Was ethical clearance obtained for their use? If so, please provide the details in the manuscript.
- The heading "Mycobacterial Infections" seems too broad given that selected Mycobacterium strains were used. A more specific heading would be appropriate.
Author Response
We sincerely appreciate the reviewer’s time and effort in thoroughly reviewing our manuscript and providing valuable constructive feedback. Thank you. Please note that, in accordance with the journal's requirements, we have moved the Materials and Methods section before the Results section. Consequently, the line numbers have changed. However, all modifications are clearly highlighted in red for ease of review.
In the manuscript titled "Discovery of Biofilm-Inhibiting Compounds to Enhance Antibiotic Effectiveness Against Mycobacterial Infections," the authors have screened a library of small molecules to identify compounds that inhibit biofilm formation in selected Mycobacterium species. This effort to screen a large compound library for potential biofilm-inhibiting agents is commendable. The study reports the identification of 17 active biofilm-inhibiting compounds from the screening, presenting promising results. However, I have several queries and suggestions for the authors to address:
1. Line 55 and 381: Does SCFM stand for Synthetic Cystic Fibrosis Medium or Cystic Fibrosis Sputum Medium? Please clarify.
A: Yes, the reviewer is right, SCFM stands for Synthetic Cystic Fibrosis Sputum Medium. We made this correction (lines 56 and 137). Thank you.
2. Line 91: A reference for bacterial amyloids should be added.
A: We added references as requested by the reviewer.
3. Statistical analyses and p-values are missing from the figures and methods sections. Please include these to strengthen the study's validity.
A: The figures now include the p values.
4. Line 169: Compounds 5 and 16 demonstrate significant biofilm inhibition but also exhibit high viability inhibition. How do the authors justify these findings?
A: Compounds 5 and 16 exhibit significant antibacterial activity by not only inhibiting bacterial viability but also demonstrating potent efficacy against pre-formed biofilms. Additionally, these compounds maintain their activity under diverse conditions, including in vitro replicative environments and intracellular states. This suggests that the compounds target a critical mechanism in NTM that is universally expressed and indispensable across all stages of bacterial growth and survival, making them promising candidates for broad-spectrum therapeutic applications. We added this statement in the discussion text (line 496).
5. Is AMK referring to an aminoglycoside class of antibiotics or specifically to Amikacin? This ambiguity leads to confusion while reading the manuscript.
A: We revised the sentence in line 470 to ensure clarity and eliminate any potential confusion for the reader.
6. Figure 5: In addition to the percentage graph, providing a CFU graph (log/CFU) would help readers better understand the extent of log reduction achieved after treatment.
A: We agree with the reviewer that CFU data can effectively demonstrate a log reduction in bacterial counts. However, due to variability in infection levels between biological replicates, particularly for the fast-growing MAB, it is more appropriate to calculate statistical significance using the percentage survival data. Otherwise, Figure 5 can represent CFU data from a single biological study, lacking standard deviations, and only report/mention in the text a statistical significance between biological replicates based on percentage survival.
7. Line 376: What conditions were used for sonication? Improper sonication can lead to bacterial killing or cell wall breakage, which might affect the results.
A: This is a water bath sonicator that we use for only 1 minute (no other settings) to resuspend bacteria. We also plate the bacteria to calculate the original inoculum, ensuring accurate inoculum concentrations. We have included this clarification in the methods section (Line 129). Thank you.
8. For Synthetic Cystic Fibrosis Medium, was the media composition formulated by the authors or adapted from previous publications? Please specify.
A: We adopted the SCFM from publication #37 as referenced in line 138.
9. Line 453: The sentence is incomplete. A thorough proofreading of the manuscript is recommended before resubmission.
A: We have revised this sentence (Line 211).
10. Line 454: Post-biofilm methods are mentioned in the heading but not described in the methods section.
A: We revised the paragraph to clarify pre- and post-biofilm condition studies (Lines 202-210). Thank you.
11. Please provide the full forms of IC50 and CC50 for clarity.
A: We made these changes (lines 228 and 241).
12. How do the compounds penetrate the cells to kill intracellular bacteria, while Amikacin does not?
A: Amikacin (AMK) is a bacteriostatic antibiotic that demonstrates concentration-dependent bactericidal activity at higher concentrations and activity correlates to bacterial concentrations as well (Biofilms are forms at 107 and IC50 studies in growth media are done with 105). Additionally, following the infection of THP-1 macrophages at 105 concentrations, we treat the cells with 200 μg/mL AMK for 2 hours to eliminate extracellular bacteria, during which it exhibits its bactericidal effect. Subsequently, the cells are treated with a 2×MIC concentration (32 μg/mL) to assess and compare compound activity alongside the antibiotic’s effectiveness against intracellular bacteria. The 2×MIC concentration is also utilized in antibiotic-compound combination experiments conducted in in vitro culture media.
13. Including CV staining images in the figures would make the manuscript more engaging for readers.
A: We appreciate the suggestion to include crystal violet staining images of plates to enhance the manuscript's engagement. However, the use of OD measurements provides a more precise and reproducible means of assessing biofilm formation compared to visual interpretation of staining images, which can be subjective. The CV staining images would not add value to the quantitative focus of our findings and for this reason, we chose to exclude images.
14. The authors have used clinical strains in this study. Was ethical clearance obtained for their use? If so, please provide the details in the manuscript.
A: The principal investigator has obtained BSL-2 approval to conduct studies on all mycobacterial BSL-2 organisms, including clinical isolates. The approval protocol number and statement have been included in the Materials and Methods section for a reference (line 122-123).
15. The heading "Mycobacterial Infections" seems too broad given that selected Mycobacterium strains were used. A more specific heading would be appropriate.
A: We changed “Mycobacterial” with M. abscessus.
Reviewer 2 Report
Comments and Suggestions for Authors
The manuscript by Danelishivili et al reports development of HTS assay for the potencial treatment of MAB, which is still a challeging issue given its ability to produce biofilms. The reviewer suggest publishing after addressing following questions:
1. Table 1. some structures are not showing properly.
2. Considering some compounds are not very stable in ambient atmosphere(eg. compound 1, 10, 16 compound 11 ). Even though these compounds were purchased from conmmercial source the author should check the purity of the powder.
3. Line 240. The treatment is not properly described.
4. Given the pre-exposure treatment seem to have better efficacy against the bacteria, can the author explain the reason why the post-exposure were used in the intracellular potency test?
Comments on the Quality of English LanguageOverall this manuscript has a good quality in English but I would suggest the author to use full name to avoid confusion, eg. NTM in line 36, CLA in line 115 and AMK in figure 1.
Author Response
1. Table 1. some structures are not showing properly.
A: We made sure that all compound structures are visible in the table. Thank you.
2. Considering some compounds are not very stable in ambient atmosphere(eg. compound 1, 10, 16 compound 11 ). Even though these compounds were purchased from conmmercial source the author should check the purity of the powder.
A: These compounds are distributed to research communities worldwide, with their purity verified by the vendors. As a microbiology lab, we lack the capacity to independently assess their purity; however, our confirmatory assays consistently yield reproducible results, ensuring the compounds consistency.
3. Line 240. The treatment is not properly described.
Post-infection in this section means that macrophages have engulfed bacteria and the pathogen is in the intracellular state (see more explanation below).
4. Given the pre-exposure treatment seem to have better efficacy against the bacteria, can the author explain the reason why the post-exposure were used in the intracellular potency test?
A: In addition to biofilms, we tested select compounds under various conditions, including the intracellular state. When MAB is introduced to phagocytic cells, these immune cells naturally engulf the bacteria, transitioning them to an intracellular state. This allows us to assess the compounds' effectiveness over time, as the bacteria will not remain extracellular in this environment.
Reviewer 3 Report
Comments and Suggestions for Authors
The manuscript "Discovery of Biofilm-Inhibiting Compounds to Enhance Antibiotic Effectiveness Against Mycobacterial Infections" is well written.
1. The Abstract and Introduction are good. In the Introduction, the authors should make short sentences to be more explicit.
2. Incorporate more organized subheadings in the "Results" section to enhance the findings, more easy. In the discourse, equilibrate technical profundity with accessibility to guarantee that the results are comprehensible to wider audiences.
3. In "4. Materials and Methods", please add a subchapter for all your bolts. (e.g. "Bacterial strains and growth culture.")
4. In "4. Materials and Methods", row 449, find a citation for the formula.
5. The statement about “Data availability” is missing.
6. The authors should provide Conclusions to summarize the entire manuscript.
7. The References should be from the past 5 years.
8. The bacterial activity should be more visible (the numbers should be larger) in Figure 2.
9. Overall, the manuscript is well written, and the English is good.
With all these considerations, I recommend a minor revision.
Author Response
The manuscript "Discovery of Biofilm-Inhibiting Compounds to Enhance Antibiotic Effectiveness Against Mycobacterial Infections" is well written.
1. The Abstract and Introduction are good. In the Introduction, the authors should make short sentences to be more explicit.
A: We improved the introduction.
2. Incorporate more organized subheadings in the "Results" section to enhance the findings, more easy. In the discourse, equilibrate technical profundity with accessibility to guarantee that the results are comprehensible to wider audiences.
A: We modified some subheadings to align with results.
3. In "4. Materials and Methods", please add a subchapter for all your bolts. (e.g. "Bacterial strains and growth culture.")
A: We added subchapter for all bolts. Thank you.
4. In "4. Materials and Methods", row 449, find a citation for the formula.
A: The biofilm percentage calculation does not require a citation, as it follows a standard percentage calculation method. In this case, the calculation is normalized using the negative control, ensuring accuracy and consistency. Referencing a research paper in this context would technically reference the research article as a whole, rather than the specific formula, which is unnecessary given the straightforward nature of the calculation.
5. The statement about “Data availability” is missing.
A: We added the statement. Thank you.
6. The authors should provide Conclusions to summarize the entire manuscript.
A: We added conclusion section. Thank you.
7. The References should be from the past 5 years.
A: The Instructions for Authors of the Pharmaceuticals journal does not specify that references must be limited to those published within the last five years.
8. The bacterial activity should be more visible (the numbers should be larger) in Figure 2.
A: We appreciate the suggestion to enlarge the compound numbers. However, due to the close proximity of several inhibition data points, increasing the font size causes the compound numbers to overlap and obscure the actual data. Therefore, we kindly request to keep the font size as it is to maintain the clarity of the plots.
9. Overall, the manuscript is well written, and the English is good.
A: We sincerely appreciate the reviewer’s time and effort in thoroughly reviewing our manuscript and providing valuable constructive feedback. Thank you.
Reviewer 4 Report
Comments and Suggestions for Authors
This article addresses an important issue by investigating the discovery of compounds effective against Mycobacterium abscessus biofilms and their potential to increase antibiotic efficacy. Although the study has a strong experimental design and a comprehensive method description, the clinical applicability and translational importance of the findings are not sufficiently emphasized. In particular, limitations such as the extent to which the artificial sputum medium used reflects natural lung conditions and the lack of in vivo studies are not addressed, and the mechanistic dimensions of the results are not discussed in depth. Although the results are clear and supported by visuals, the presentation of statistical analyses and error bars should be clarified. Other elements that limit the scientific contribution of the article include the lack of concrete recommendations for future studies and the lack of a clear definition of the literature gap. Nevertheless, the study provides a promising basis for the discovery of antibiofilm compounds.
I have a few suggestions regarding the article below.
1. Title: It could be more specific. For example, an emphasis could be placed on the chemical classes or mechanisms investigated in the study.
2- Abstract: It would be useful to present the findings in a more quantitative manner. For example, instead of “17 hits identified”, the percentage activities of the most effective compounds should be stated.
2. Introduction: The literature gap needs to be defined more clearly. For example, the question “What is the novelty of this study?” is not fully answered. A critical analysis of existing studies can be added to clarify the difference of this study with alternative approaches.
3. Materials and Methods:
Limitations of the experiments performed are not stated. For example, it is debatable how well the artificial sputum medium used reflects the natural lung microenvironment.
Statistical analysis methods should be clearly stated and the software used should be stated.
4. Results: The clinical meaning of the results is not clearly explained. For example, the question “How close are these compounds to human use?” should be addressed. In some graphs (e.g. Figures 2 and 4), the meaning and statistical significance of the error bars are unclear. They should be presented more clearly.
5. Discussion: The discussion section falls short in explaining the potential mechanisms of the findings. For example, a molecular level explanation is not provided as to why the selected compounds are effective.
The limitations of the study should be clearly stated. For example, the lack of in vivo experiments can be considered a major limitation.
6. Conclusion and Future Work: How the findings of the study can be transferred to practice should be discussed more clearly. For example, it can be mentioned which steps are required for the transition to clinical trials. More concrete suggestions can be provided for future studies.
Author Response
We sincerely appreciate the reviewer’s time and effort in thoroughly reviewing our manuscript and providing valuable constructive feedback. Thank you.
I have a few suggestions regarding the article below.
- Title: It could be more specific. For example, an emphasis could be placed on the chemical classes or mechanisms investigated in the study.
A: Because most identified compounds belong to diverse chemical classes, it is challenging to highlight any specific class in the title. Additionally, our study did not focus on elucidating the molecular mechanisms of compound activity. Our primary aim was to discover compounds with biofilm inhibitory activity and then test on other phenotypes such as intracellular effects, and synergistic properties. Investigating the mechanisms of action will be an objective for future research.
2- Abstract: It would be useful to present the findings in a more quantitative manner. For example, instead of “17 hits identified”, the percentage activities of the most effective compounds should be stated.
A: We have incorporated the reviewer’s suggestion in the abstract.
- Introduction: The literature gap needs to be defined more clearly. For example, the question “What is the novelty of this study?” is not fully answered. A critical analysis of existing studies can be added to clarify the difference of this study with alternative approaches.
A: While numerous high-throughput compound screening assays have been conducted, leading to the identification of various classes of antibacterial compounds through the use of in vitro grown mycobacteria in diverse culture media and conditions, this approach often overlooks compounds that might target bacterial factors exclusively expressed during the physiological stages of biofilm formation or during maintenance of biofilm integrity. Consequently, environments that mimic in vivo conditions such as synthetic cystic fibrosis sputum media represent more biologically relevant conditions for drug discovery, compared to standard growth media or buffers. These more complex environments potentially may reveal compounds with novel mechanisms of action that target bacterial pathogenicity factors essential for the integrity of non-tuberculous mycobacterial biofilms. Therefore, in our HTS assay we employed artificial sputum media to simulate the CF lung environment, providing a better model for biofilm-associated NTM infections. We clarified this information better in the introduction.
- Materials and Methods: Limitations of the experiments performed are not stated. For example, it is debatable how well the artificial sputum medium used reflects the natural lung microenvironment. Statistical analysis methods should be clearly stated and the software used should be stated.
A: While the SCFM is not an exact replica of CF patient sputum, it is a widely accepted model for studying the behavior and pathogenicity of NTM pathogens as well as Pseudomonas species. Additionally, we have included a section on statistical analysis, specifying the software used. Thank you.
- Results: The clinical meaning of the results is not clearly explained. For example, the question “How close are these compounds to human use?” should be addressed. In some graphs (e.g. Figures 2 and 4), the meaning and statistical significance of the error bars are unclear. They should be presented more clearly.
A: While some compounds demonstrate potency in vitro, it is challenging to assess their draggability at this stage without in vivo studies. Many of the compounds would likely need to undergo chemical modifications to enhance their activity while minimizing toxicity. Notably, certain compounds, as highlighted in the paper, exhibited increased toxicity when tested in phagocytic cells compared to bacterial infection models. These findings underscore the importance of in vivo studies to better understand the clinical relevance and safety of these compounds before advancing to human use. We incorporated this statement in the text.
The updated graphs now include error bars, providing a clearer representation of the variability and statistical significance of the data.
- Discussion: The discussion section falls short in explaining the potential mechanisms of the findings. For example, a molecular level explanation is not provided as to why the selected compounds are effective.
The limitations of the study should be clearly stated. For example, the lack of in vivo experiments can be considered a major limitation.
A: The primary aim of this study was to identify compounds with anti-NTM biofilm activity through high-throughput screening of a diverse compound library in an environment mimicking CF patients’ sputum. We successfully identified 17 compounds from varied functional classes, with compounds 5 and 16 demonstrating exceptional antibacterial activity. These compounds not only inhibited bacterial viability but also effectively targeted pre-formed biofilms, including exhibiting potency in in vitro replicative environments and intracellular states. These findings suggest that the selected compounds act on a critical, universally expressed mechanism in NTM, which is central across all stages of bacterial growth and survival. This positions them as good candidates for broad-spectrum therapeutic applications. However, the precise molecular mechanisms underlying their effectiveness remain unclear and require further investigation using molecular tools. We have added this clarification to the discussion section.
Regarding limitations, we acknowledge that the lack of in vivo studies is a significant constraint. While the compounds show promise, their druggability at this stage cannot be fully assessed without evaluating their performance in animal models. Also, some chemical modifications may be necessary to optimize their activity and minimize toxicity for future therapeutic use. These limitations are now clearly addressed in the revised text.
- Conclusion and Future Work: How the findings of the study can be transferred to practice should be discussed more clearly. For example, it can be mentioned which steps are required for the transition to clinical trials. More concrete suggestions can be provided for future studies.
A: We added the conclusion section as requested by the reviewer.
Round 2
Reviewer 1 Report
Comments and Suggestions for Authors
Accept in present form
Reviewer 2 Report
Comments and Suggestions for Authors
The corresponding author is not clearly indicated. Except that, the questions from the reviewer have been answered to satisfaction. I support the publication of this manuscript.
Reviewer 4 Report
Comments and Suggestions for Authors
The authors have made the corrections by taking the suggestions into consideration. It is suitable for publication in its current form.